# 4-Pyridoxic Acid/Pyridoxine Ratio in Patients with Type 2 Diabetes is Related to Global Cardiovascular Risk Scores

**DOI:** 10.3390/diagnostics9010028

**Published:** 2019-03-06

**Authors:** Rima Obeid, Juergen Geisel, Wilfred A. Nix

**Affiliations:** 1Department of Clinical Chemistry and Laboratory Medicine, Saarland University Hospital, Building 57, 66424 Homburg/Saar, Germany; juergen.geisel@uks.eu; 2Department of Neurology, Mains University Hospital, Langenbeckstr 1, 55131 Mainz, Germany; nix@arztkolleg.de

**Keywords:** diabetes, vitamin deficiency, vitamin B6 catabolism, vascular risk, 4-pyridoxic acid

## Abstract

Background: Vascular diseases are multifactorial and several risk factors may have synergetic effect on the global vascular risk. Among patients with diabetes, we investigated whether vitamin B6 species differ according to global cardiovascular risk. Methods: The present observational study included 122 patients with type 2 diabetes (mean (SD) age = 69.9 (9.1) years; 50% men). Concentrations of vitamin B6 vitamers were measured. Classical blood biomarkers and risk factors were used to compute a multivariate risk score. Results: Plasma concentrations of 4-pyridoxic acid were higher in patients with high risk versus those with low risk scores (48.2 (63.7) vs. 31.9 (15.0) nmol/L; *p* = 0.031). Plasma pyridoxine was significantly lowered in patients at high risk (2.8 (28.4) vs. 38.1 (127.8) nmol/L; *p* = 0.003). PAr index (4-pyridoxic acid/pyridoxal + pyridoxal 5′-phosphate) (1.05 (0.07) vs. 0.84 (0.06); *p* = 0.017) and the ratio of 4-pyridoxic acid/pyridoxine (7.0 (4.8) vs. 3.9 (3.2); *p* < 0.001) were higher in patients at high risk. After adjustment for cystatin C and C-reactive protein, only pyridoxine and 4-pyridoxic acid/pyridoxine ratio remained significantly different according to vascular risk scores. 4-Pyridoxic acid/pyridoxine ratio was the best marker to discriminate between patients according to their risk scores—area under the curve (AUC) (95% confidence intervals (CI)) = 0.72 (0.62–0.81). 4-Pyridoxic acid/pyridoxine ratio was directly related to plasma levels of soluble vascular cell adhesion molecule 1. Conclusion: Vitamin B6 metabolism was shifted in patients with multiple vascular risk factors. The catabolism to 4-pyridoxic acid was enhanced, whereas the catabolism to pyridoxine was lowered. High 4-Pyridoxic acid/pyridoxine ratio is independently associated with global cardiovascular risk.

## 1. Introduction

Accelerated vitamin B6 catabolism is associated with poor clinical outcomes in epidemiological studies on vascular diseases or cancer [1,2]. Plasma pyridoxal 5′-phosphate (PLP) (i.e., the active vitamin B6 form) shows a dose–response inverse association with chronic inflammation and the presence of vascular disease [3,4]. The relationship between plasma PLP levels and vascular disease could be partly due to inflammation [3]. Vitamin B6 participates as a cofactor in producing metabolites of immunomodulatory functions such as kynurenines [5].

Concentrations of PLP (commonly measured in routine diagnosis of vitamin B6 deficiency) may not mirror an underlying disturbance in vitamin B6 vitamers (Appendix A, Appendix A). The ratio of 4-pyridoxic acid/(pyridoxal + pyridoxal 5′-phosphate), also called PAr index, is a marker of enhanced vitamin B6 catabolism especially during inflammation [6,7]. PAr is markedly elevated in patients with renal insufficiency [8]. This ratio shows a high within-person-reproducibility that enables using it as an index of vitamin B6 catabolism and an atherosclerosis risk marker [6]. Elevated PAr index is associated with a significant increase in the risk of future endpoints such as stroke [9] or mortality [2]. In the Hordaland Health Study, PAr has been shown to be a significant predictor of the risk of stroke in strata of age, sex, body mass index (BMI), smoking, hypertension, and diabetes [9]. Moreover, PAr was the strongest predictor of future stroke compared with each of the classical risk factors (diabetes, smoking, hypertension, glomerular filtration rate (GFR), and C-reactive protein (CRP)) [9]. However, vascular diseases including stroke are multifactorial and the presence of several risk factors in one person increases the risk of future vascular events compared with the presence of any of the risk factors alone. It is not clear if dysregulations of vitamin B6 metabolism (i.e., measured using the surrogate marker PAr index) can be predicted by the presence of several traditional vascular risk factors or with elevated plasma concentrations of early markers of atherosclerosis such as soluble vascular cell adhesion molecule 1 (sVCAM-1).

The association between concentrations of vitamin B6 forms and global vascular risk deserves further investigation. In patients with type 2 diabetes who are at increased risk for atherosclerosis, we hypothesized that dysregulation of vitamin B6 metabolism shows a dose–response relationship with composite vascular risk scores that are used as surrogate markers of future atherosclerosis risk. We computed vascular risk scores from a number of traditional risk factors such as hypertension, hyperlipidemia, age, and smoking. In addition, we studied the distribution of plasma and urine vitamin B6 forms according to the multivariate cardiovascular risk scores and studied the association between vitamin B6 forms and plasma sVCAM-1 levels in patients with diabetes.

## 2. Materials and Methods

### 2.1. Study Population and Setting

Consecutive patients with type 2 diabetes were recruited to an observational study that aimed to investigate the association between incipient nephropathy and vitamin markers in patients with type 2 diabetes [10]. In total, 122 participants were recruited at three general and internal medicine practices in Germany between January and September 2011. During the routine physician visits, measurements of blood pressure, heart rate, and body mass index (BMI) were performed and blood and urine samples were collected. Moreover, we used a standardized questionnaire to collect information on medical history and risk factors. Medical records were reviewed for disease history.

Inclusion criteria were the following: Adult men and women, type 2 diabetes mellitus for five years or longer, and a BMI between 19 and 40 kg/m^2^. Exclusion criteria were the following: Patients with a creatinine clearance of less than 10 mL/min, high alcohol consumption (>50 U/week), serious comorbidities such as cancer, renal transplantation, use of any vitamin B supplement including multivitamin preparations within one month prior to recruitment, and pregnant and lactating women. A group of 34 apparently healthy staff subjects (age ≥35–65 years; mean age 49 years) who were free of diabetes was recruited from the same centers during the same time as the patients. The study is registered at ClinicalTrial.gov. ID: NCT03447275. The study protocol was reviewed and approved (approval number: 837.320.10/7329, approval date: 22 January 2010) by the regional ethical committee (Landesärztekammer Rheinland-Pfalz), Mainz, Germany. The study was conducted in agreement with the ethical Principles for Medical Research Involving Human Subjects as documented in Helsinki declaration. All participants provided a written informed consent to the study.

### 2.2. Exposure Definition

The main exposure in the current study is a multivariate cardiovascular risk score that has been developed and validated in patients with diabetes [11]. This score is computed from common clinical measurements available in primary care settings, such as age, sex, BMI, total cholesterol, high density lipoprotein cholesterol (HDL-C), low density lipoprotein cholesterol (LDL-C), systolic blood pressure, diastolic blood pressure, and HbA1c (Appendix A, Appendix A). The score estimates the two-year risk of hospitalization due to a cardiovascular event in patients with type 2 diabetes [11]. The present observational study used already established global risk scores and it was not planned to be prospective or to prove if subjects would attain the outcome predicted by the risk scores in the future.

To prove the validity of our hypothesis, we also computed two other cardiovascular risk scores that estimate the long-term likelihood of a person (age range between 20 years and 75 years) to develop a cardiovascular event within 10 years based on his/her risk factors. The Prospective Cardiovascular Münster (PROCAM) score has been developed in a large German cohort and it estimates the 10-year risk of a major coronary event [12]. The PROCAM score is computed from the following classical risk factors: the presence of type 2 diabetes, sex, and smoking (yes vs. no) as categorical factors and plasma concentrations of LDL-C, triglycerides, HDL-C, and systolic blood pressure as continuous variables [12]. Moreover, the U.S. Framingham score was computed from 93 out of the 122 participants with type 2 diabetes from the following risk factors: the presence of type 2 diabetes, sex, and smoking as categorical variables and total cholesterol, HDL-C, age, and systolic and diastolic blood pressure as continuous variables. The PROCAM and Framingham scores were not calculated from 29 patients with diabetes (18 men, 11 women) who were older than 75 years. The PROCAM and Framingham scores were computed from the 34 apparently healthy subjects.

### 2.3. Study Outcomes

The primary outcomes among patients with diabetes were differences between two exposure levels (high and low vascular risk) in plasma or urine concentrations of vitamin B6 forms (4-pyridoxic acid, pyridoxal, PLP, pyridoxamine, pyridoxamine phosphate, and pyridoxine), PAr index, and 4-pyridoxic acid / pyridoxine ratio.

The secondary outcomes of the study were differences in plasma concentrations of sVCAM-1 and CRP among patients with diabetes between those with high and low vascular risk. A secondary aim was to investigate a possible association between vitamin B6 forms and vascular risk among the control subjects who had generally lower global risk than the patients with diabetes (Appendix A, Appendix A). The study did not intend to compare vitamin B6 forms between patients with diabetes and the control subjects.

### 2.4. Biological Samples and Analytical Methods

Blood samples were collected in the morning after an overnight fast. Blood samples were collected into tubes containing K^+^EDTA and in tubes without anticoagulant (10 mL). EDTA plasma and serum were separated immediately from blood cells by centrifugation for 10 min at 2000× *g* at ambient temperature; aliquots were snap-frozen and stored at −80 °C until analysis. On the morning of blood collection, a sample of spontaneous urine was collected and kept as aliquots at −80 °C until measuring urinary markers. Analytical methods were conducted in two central laboratories: Central Laboratory of the University of Warwick, U.K. (vitamin B6 forms and sVCAM-1) and Central Laboratory of the Saarland University Hospital, Germany (clinical routine markers such as plasma glucose, liver enzymes, CRP, creatinine, cystatin C, and lipids).

The following vitamin B6 forms were measured in plasma and urine: PLP, pyridoxamine phosphate, pyridoxine, pyridoxamine, pyridoxal, and 4-pyridoxic acid. Hiigh-performance liquid chromatography (HPLC) (UltiMate^®^ 3000, Thermo Scientific, Dionex, UK) and fluorometric detection was used to measure vitamin B6 forms as reported elsewhere [13]. Fasting plasma glucose (HemoCue Glucose 201+ analyzer, HemoCue AB, Willich, Germany), blood HbA1c (high-performance liquid chromatography, immunoturbidimetry) and serum creatinine (Jaffé photometric method) were measured using automated methods. The glomerular filtration rate (GFR) was estimated according to the Modification of Diet in Renal Disease formula. Serum total cholesterol, triglyceride, LDL-C, and HDL-C concentrations were measured by using automated enzymatic colorimetric assays. Plasma concentrations of alanine aminotransferase (ALT) and aspartate aminotransferase (AST) were determined using a standardized UV kinetic method (cobas^®^ system, Roche Diagnostics GmbH, Mannheim, Germany). Serum concentrations of sVCAM-1, cystatin C and CRP were analyzed by quantitative sandwich enzyme immunoassays (Quantikine^®^ Colorimetric Sandwich ELISA Kits, R&D Systems, Abingdon, UK) according to the manufacturer’s instructions.

### 2.5. Statistical Analysis

Statistical analyses were conducted using BM SPSS^®^-Statistics software (Version 25, IBM Corp., Armonk, NY, USA). Kolmogorov–Smirnov test and Lilliefors significance correction were applied to test the distribution of the data. All variables were not normally distributed except for serum cholesterol, LDL-C, and the vascular risk scores. Continuous variables are shown as mean and standard deviation (SD) in addition to median [interquartile range] for the main markers. The non-parametric Mann-Whitney test (for continuous variables) and the chi-squared test (for categorical variables) were used to compare the markers among patients with diabetes according to the risk score category. Adjustment for possible confounders was conducted by running a univariate analysis of variance test (based on ANOVA test) on the log-transformed variables, where patients with low versus those with high risk scores were compared and cystatin C or cystatin C and CRP were entered as a covariate to adjust for renal function and inflammation, respectively. The correlations between different variables were tested using the Spearman test.

The receiver operating curve (ROC) analysis was conducted to compare the markers (4-pyridoxic acid/pyridoxine ratio, PAr index, sVCAM-1, creatinine, cystatin C, and CRP) in their discrimination power between patients at low risk and those at high risk for cardiovascular hospitalization. The area under the curve (AUC) and 95% confidence intervals (CI) of AUC were calculated for the candidate markers. Finally, *p*-values of <0.05 were considered statistically significant and those between 0.05 and 0.10 were considered to indicate a tendency.

## 3. Results

### 3.1. Population Characteristics

The study included 122 patients with type 2 diabetes (mean age 69.9 years, 50% men). The mean (SD) duration of diabetes was 11.4 (5.6) years. Table 1 shows the main characteristics and risk factors in the whole group and in patient subgroups according to the risk of cardiovascular hospitalization. The risk score was calculated for each patient based on routine clinical variables (Appendix A, Appendix A) [11]. The risk of cardiovascular hospitalization (= 1/(1 + e^−risk score^)) ranged from 1.6 to 21.3. The median of the risk in the whole group was used to stratify patients into 2 groups (high and low risk). The risk of cardiovascular hospitalization was (mean (SD) = 4.1 (1.1)) in the low-risk group and 9.8 (3.4) in the high-risk group.

The differences in main risk factors according to the risk of cardiovascular hospitalization among patients with diabetes are presented in Table 1. Compared with patients in the low-risk group, those in the high-risk category were older, had higher systolic blood pressure, serum creatinine, and cystatin C, slightly elevated CRP, elevated sVCAM-1, and lowered GFR, total cholesterol, and LDL-C. 

The control group included 34 subjects who were younger and at low risk for atherosclerosis compared with the patients with diabetes (characteristics of this control group are shown in Appendix A, Appendix A).

### 3.2. Plasma Vitamin B6 Species in Patients with Type 2 Diabetes According to Cardiovascular Hospitalization Risk

The plasma concentrations of vitamin B6 forms are shown in Table 2 in the whole group and in subgroups according to risk scores. 4-Pyridoxic acid was higher in patients in the high-risk group compared to those in the low-risk group (mean = 48.2 vs. 31.9 nmol/L: *p* = 0.031). The differences in concentrations of 4-pyridoxic acid was not significant between the two risk categories after adjusting for serum cystatin C (*p* = 0.168) or for cystatin C and CRP (*p* = 0.166). The percentage of plasma 4-pyridoxic acid of total vitamin B6 (i.e., the sum of vitamin B6 forms) was higher in the high-risk group, even after adjustment for cystatin C and CRP (*p* = 0.005). Plasma concentration of pyridoxine was significantly lower in the high-risk group compared to that in the low-risk group (12.8 vs. 38.1 nmol/L; *p* = 0.003). This difference remained significant after adjustment for cystatin C (*p* = 0.007) or for cystatin C and CRP (p = 0.010). Plasma concentrations of pyridoxal, PLP, pyridoxamine, and pyridoxamine phosphate, or the sum of all vitamin B6 forms did not differ significantly between the risk categories. Thus, 4-pyridoxic acid (elevated by 51%) and pyridoxine (lowered by 66%) were the vitamin B6 forms with marked changes in patients with high vascular risk scores compared to those with low risk scores.

The ratio of 4-pyridoxic acid/pyridoxine was markedly higher in the high-risk group (7.0 vs. 3.9, differ by 79%, *p* < 0.001), mirroring the shift in vitamin B6 metabolism toward more 4-pyridoxic acid and less pyridoxine in patients with high risk. The difference in 4-pyridoxic acid/pyridoxine ratio between the high- and the low-risk categories remained significant after adjustment for cystatin C (*p* < 0.001) or for cystatin C and CRP (*p* = 0.001).

The PAr index was significantly higher in the high-risk groups compared with the low-risk group (differs by 28%, *p* = 0.017), but this difference was ameliorated after adjustment for cystatin C (*p* = 0.117) or cystatin C and CRP (*p* = 0.282).

In contrast to PAr, plasma 4-pyridoxic acid/pyridoxine ratio was significantly higher in patients with higher PROCAM score (10-year risk of a major vascular event), even after adjustment for cystatin C and CRP (Appendix A, Appendix A). PAr (*p* = 0.234) and 4-pyridoxic acid/pyridoxine ratio (*p* = 0.843) did not differ between patients with diabetes according to the Framingham score divided by the median of the whole group (available from 93 patients). Thus, compared to PAr index, the ratio of 4-pyridoxic acid/ pyridoxine showed a stronger elevation in patients with high risk scores compared to those with low risk scores.

### 3.3. Urinary Vitamin B6 Species in Patients with Type 2 Diabetes According to Cardiovascular Hospitalization Risk

Urinary concentrations of 4-pyridoxic acid (expressed as µmol/g creatinine) showed no significant differences between the high- and the low-risk groups (Table 3). Urinary PAr index (*p* = 0.041) and 4-pyridoxic acid/pyridoxine ratio (*p* = 0.022) were significantly higher in the high-risk group as compared with the low-risk group. These differences were in the same direction as for the corresponding plasma markers. However, the differences in urinary PAr and 4-pyridoxic acid/pyridoxine ratio showed only a tendency to be higher in the high-risk group after adjustment for cystatin C or cystatin C and CRP. Therefore, the lowered mean plasma pyridoxine in patients with diabetes and high-risk scores (Table 2) cannot be explained by increased excretion of this form in urine (Table 3).

### 3.4. Correlations and ROC-Curve Analyses in Patients with Type 2 Diabetes

Concentrations of 4-pyridoxic acid and PLP in plasma showed a direct moderate correlation (Spearman coefficient of correlation r = 0.521; *p* < 0.001). Concentrations of 4-pyridoxic acid in plasma and those in urine were positively correlated (r = 0.467; *p* < 0.001). Moreover, plasma and urine concentrations of pyridoxine did not correlate (r = 0.063; *p* = 0.500). Thus, urinary excretion (i.e., renal dysfunction) could modify plasma 4-pyridoxic acid, whereas plasma pyridoxine might be seen as independent on renal function, and influenced by endogenous vitamin B6 metabolism.

We used ROC-curve to investigate which of the following markers—4-pyridoxic acid/pyridoxine ratio, PAr, sVCAM-1, renal function markers (cystatin C, creatinine), or CRP—could better discriminate between patients with high and those with low cardiovascular hospitalization risk (Table 4). The area under the curve (AUC) was highest for 4-pyridoxic acid/pyridoxine ratio (AUC = 0.717; *p* < 0.001) followed by sVCAM-1 (AUC = 0.681; *p* = 0.001).

Serum concentrations of CRP and sVCAM-1 showed differential patterns of correlations with plasma vitamin B6 forms. In patients with low risk, serum CRP was directly correlated with pyridoxine phosphate (r = 0.403, *p* = 0.001). In patients with high risk, serum sVCAM-1 was directly associated with plasma 4-pyridoxic acid (r = 0.256, *p* = 0.046) and PAr ratio (r = 0.510, *p* < 0.001) and negatively correlated with plasma pyridoxal (r = −0.322, *p* = 0.011).

Appendix A (see Appendix A) shows that PAr index among patients with diabetes is correlated with age, renal function markers, CRP as an inflammation marker, and the risk score according to Yu et al. [11]. On the other hand, the 4-pyridoxic acid/pyridoxine ratio showed significant correlations with age, sVCAM-1 (as an early marker for atherogenesis), CRP, and the risk scores (Yu et al. and PROCAM). Thus, in patients with type 2 diabetes, the PAr index and 4-pyridoxic acid/pyridoxine ratio showed a differential relationship to atherosclerosis, renal dysfunction, and inflammation.

### 3.5. Vitamin B6 Forms in Subjects without Diabetes

Appendix A (see Appendix A) shows that both PAr index and 4-pyridoxic acid/pyridoxine ratio showed significant differences between diabetes-free subjects and the 61 patients with diabetes who were at low vascular risk. On the other hand, plasma levels of 4-pyridoxic acid and pyridoxine did not differ significantly between these two groups. PROCAM and Framingham scores were generally low in the subjects without diabetes. There were no significant correlations between PAr index or 4-pyridoxic acid/pyridoxine ratio and PROCAM or Framingham scores among the diabetes-free individuals.

## 4. Discussion

The present study has shown enhanced vitamin B6 catabolism to 4-pyridoxic acid in plasma of patients with diabetes and high vascular risk scores compared to patients at low risk. Moreover, levels of plasma pyridoxine were low, whereas levels of plasma PLP (i.e., the classical vitamin B6 marker) did not differ between patients with high risk and those with low risk. The shift in vitamin B6 metabolism in patients with high risk was best reflected by a markedly elevated 4-pyridoxic acid/pyridoxine ratio (7.0 vs. 3.9). The 4-pyridoxic acid/pyridoxine ratio was a strong positive predictor of vascular risk scores. Elevated 4-pyridoxic acid/pyridoxine ratio appeared to reflect the presence of atherosclerosis independent of renal function and inflammation. In addition, PAr index, which is known to be elevated in inflammation, was higher in patients at high vascular risk compared with those at low risk. Elevation of PAr in patients with diabetes was mostly explained by levels of renal function markers and CRP. In contrast, 4-pyridoxic acid/pyridoxine ratio showed a significant relationship with the atherosclerotic marker sVCAM-1 and only weak or no association with inflammatory and renal function markers. Overall, in patients with diabetes, we observed a link between global vascular risk and a shift in plasma vitamin B6 forms that favored catabolism of vitamin B6 to 4-pyridoxic acid at the expense of pyridoxine.

The present observational study suggests that elevated plasma 4-pyridoxic acid levels could be a surrogate marker that reflects the presence of other risk factors. Our results are consistent with earlier studies showing an association between plasma 4-pyridoxic acid and renal insufficiency or inflammation [2]. Plasma 4-pyridoxic acid predicted long-term mortality in patients with angina pectoris, but the associations were weakened after correcting for markers of inflammation in plasma [2]. In line with our study, plasma 4-pyridoxic acid has been shown to discriminate between patients with confirmed coronary heart disease as compared with controls who were free of coronary heart disease in a study using a non-targeted metabolomics approach [14].

4-Pyridoxic acid is formed from pyridoxal by aldehyde oxidase (AOX) in the liver (Appendix A, Appendix A). The excess of 4-pyridoxic acid is rapidly excreted in urine. Urinary concentrations of 4-pyridoxic acid increase rapidly after supplementing vitamin B6 [15] and they show a dose–response relationship with vitamin B6 intake [16]. Under a stable vitamin B6 intake, elevated plasma 4-pyridoxic acid could be partly due to renal insufficiency. On the other hand, increased catabolism of PLP to 4-pyridoxic acid via pyridoxal (i.e., increased PAr index) could be due to enhanced AOX activity that produces hydrogen peroxide and may contribute to nitric oxide formation [17].

Elevated plasma PAr index has been shown to be associated with a significant increase in the risk of incident stroke in a population-based study [9] or with the risk of future mortality in patients with angina pectoris [2]. The association between PAr and clinical outcomes remained significant after multiple adjustments [2,9]. Compared with 4-pyridoxic acid, PAr offers a better prognostic value because it identifies a shift in plasma vitamin B6 forms. Our results on differences in PAr according to the global cardiovascular risk are in line with the literature. However, in our study, the differences in PAr according to the global risk scores were mainly explained by renal dysfunction and inflammation (Table 2). Compared with available studies that documented hard endpoints such as death or objectively diagnosed vascular disease (by using angiography), our results could have been influenced by a lower degree of certainty in discriminating between patients at high and those at low risk based on classical risk factors (i.e., a probabilistic model).

Causes of the shift in vitamin B6 metabolism in patients at risk for vascular diseases are currently not clear. Wilson and Davis suggested that vitamin B6 deficiency could arise under conditions of oxidation or phosphorylation mechanisms in vitamin B6 pathway [18]. The mechanisms could be related to the role of vitamin B6 in inflammation, amino acid metabolism, or oxidative stress. Our study has shown that enhanced vitamin B6 catabolism is related to endothelial cell adhesion marker (sVCAM-1), but less to inflammatory markers.

We identified the ratio of 4-pyridoxic acid/pyridoxine as a novel marker to discriminate between low-risk and high-risk patients (highest AUC compared with PAr, or renal function and inflammatory markers). 4-Pyridoxic acid/pyridoxine ratio indicates the distribution of pyridoxal 5′-phosphate between 4-pyridoxic acid or pyridoxine. Pyridoxine could be considered a surrogate marker of vitamin B6 intake. Elevated 4-pyridoxic acid/pyridoxine ratios could in theory be due to low vitamin B6 intake, renal dysfunction, or both. The prognostic value of this ratio in cardiovascular endpoints has not been studied, but deserves further investigation. Measuring vitamin B6 species could help a patient’s stratification into “low risk” or “high risk” patients in order to tailor cardiovascular intervention or prevention. However, the clinically relevant threshold, the modifiability (i.e., upon pyridoxine supplementation), and the utility of this marker in clinical practice are not known.

Our study raises questions regarding a possible causality of vitamin B6 catabolism or low intake of pyridoxine in vascular diseases. An open question is whether the changes in vitamin B6 vitamers are reversible. Low dietary vitamin B6 intake is associated with a higher cardiovascular risk [19], but the effect of long-term replacement on health outcomes has not been investigated using vitamin B6 alone. Measuring plasma and urine vitamers after vitamin B6 load could be useful in detecting early disturbances in vitamin B6 metabolism [20], but this has to be proven. Moreover, it remains to be tested whether 4-pyridoxic acid/pyridoxine ratio or PAr index may be lowered by AOX inhibitors [21] or pyridoxine. Pyridoxine supplementation (40 mg/day for 28 days) increased concentrations of all vitamin B6 vitamers, whereas CRP remained unchanged [22]. Interestingly, vitamin B6 vitamers (i.e., PLP and 4-pyridoxic acid) showed less increase in subjects with elevated CRP [22], which could be due to increased demand for pyridoxine in inflammation. Finally, if vitamin B6 replacement may have a differential effect on vitamin B6 metabolism according to global vascular risk (i.e., different demands), then personalized supplementation may be necessary for high-risk patients.

The present study has some limitations due to its observational design, small sample size, and the absence of an age- and sex-matched group of patients without diabetes. However, our primary aim was to investigate a possible dose–response relationship between vitamin B6 forms and composite vascular risk scores among patients with diabetes who are generally at increased vascular risk. The study was explorative and an a priori sample size calculation could not be based on similar studies. Post-hoc power analysis has shown that the observed difference in 4-pyridoxic acid/pyridoxine ratio between diabetics with high and low risk (mean (SD) = 7.0 (4.8) vs. 3.9 (3.2); *n* = 122) provided 0.99 power at α = 0.05 (2-sided *t*-test) and *n* = 61 subjects in each group.

## 5. Conclusions

In patients with type 2 diabetes, 4-pyridoxic acid/pyridoxine ratio showed an independent direct association with sVCAM-1 and global cardiovascular risk scores that were based on routinely measured clinical risk factors. Plasma and urine 4-pyridoxic acid and PAr index were higher among patients with diabetes with advanced vascular risk, but the associations were mostly explained by renal function and inflammation. The shift in vitamin B6 metabolism may predict an ongoing vascular damage as reflected by sVCAM-1 and traditional cardiovascular risk factors. The relationship of vitamin B6 metabolism with global vascular risk in s non-diabetic population and the influence of possible modifying factors on 4-pyridoxic acid metabolism and clinical endpoints deserve further investigation.

## Figures and Tables

**Table 1 diagnostics-09-00028-t001:** Main characteristics of patients with type 2 diabetes according to the risk of cardiovascular hospitalization.

Risk Factors	All Patients *n* = 122	Low Risk ^1^ (<−2.8) *n* = 61	High Risk (≥−2.8) *n* = 61	*p* ^3^
Risk score ^1^, mean (SD)	−2.7 (0.6)	−3.2 (0.3)	−2.3 (0.4)	-
Risk of CVD hospitalization ^2^				-
Mean (SD)	7.0 (3.8)	4.1 (1.1)	9.8 (3.4)	
Min.–Max.	1.6–21.3	1.6–5.7	5.8–21.3	
Men, n (%)	60 (50%)	29 (48%)	32 (52%)	0.587 ^4^
Age, years	69.9 (9.1)	65 (9)	75 (5)	<0.001
Diabetes duration, years	11.4 (5.6)	10.7 (4.0)	12.2 (6.8)	0.441
BMI, kg/m^2^	29.8 (4.1)	29.9 (4.4)	29.6 (3.8)	0.751
Systolic BP, mmHg	138 (15)	135 (14)	142 (14)	0.004
Diastolic BP, mmHg	82 (7)	81 (7)	82 (7)	0.135
Heart rate, bpm	71 (9)	71 (8)	70 (9)	0.656
Creatinine, mg/dL	0.92 (0.24)	0.86 (0.19)	0.99 (0.27)	0.011
Cystatin C, µg/mL	1.03 (0.46)	0.90 (0.35)	1.15 (0.52)	0.009
GFR, mL/min	77.1 (25.5)	85 (27)	69 (21)	<0.001
Plasma glucose, mg/dL	144 (44)	142 (47)	146 (42)	0.577
HbA1c, %	7.4 (1.2)	7.5 (1.4)	7.3 (0.9)	0.699
CRP, µg/mL	5.7 (12.5)	3.2 (4.7)	8.1 (16.8)	0.071
sVCAM-1, ng/mL	531 (236)	455 (162)	607 (272)	<0.001
ALT, U/L	16 (8)	17 (8)	15 (7)	0.267
AST, U/L	27 (9)	25 (7)	28 (11)	0.294
AST/ALT (De-Ritis ratio)	1.9 (0.8)	1.7 (0.5)	2.1 (1.0)	0.013
Total cholesterol, mg/dL	197 (37)	212 (35)	181 (33)	<0.001
HDL-C, mg/dL	49 (14)	47 (16)	49 (13)	0.264
LDL-C, mg/dL	125 (34)	139 (33)	111 (30)	<0.001
Triglycerides, mg/dL	155 (81)	167 (85)	140 (69)	0.061
Microalbuminuria, n (%)	47 (38.5%)	19 (32%)	28 (46%)	0.108 ^4^
Smoking				0.115 ^4^
Never	95 (78%)	43 (72%)	51 (84%)	
Former	17 (14%)	9 (15%)	8 (13%)	
Current	10 (8%)	8 (13%)	2 (3%)	
Alcohol				0.094 ^4^
No	87 (71%)	39 (65%)	48 (79%)	
Moderate (<50 U/week)	35 (29%)	21 (35%)	13 (21%)	
Statin use, n (%)	46 (38%)	20 (33%)	25 (41%)	0.453 ^4^
Insulin use	38 (31%)	17 (28%)	21 (34%)	0.605 ^4^
Metformin use	58 (48%)	31 (52%)	26 (43%)	
No glucose lowering drug	26 (21%)	12 (20%)	14 (23%)	

Data are mean (SD), unless otherwise specified. ^1^ The risk score was calculated using the formula in Appendix A, Appendix A [11]. ^2^ The risk for cardiovascular disease (CVD) hospitalization = 1/(1 + e^−risk score^). ^3^
*p*-values are according to the Mann–Whitney test. ^4^
*p*-value is according to the chi-squared test. ALT, alanine aminotransferase; AST, aspartate aminotransferase; BMI, body mass index; BP, blood pressure; CRP, C-reactive protein; GFR, glomerular filtration rate; HbA1c, glycated hemoglobin; HDL-C, high density lipoprotein cholesterol; LDL-C, low density lipoprotein cholesterol; sVCAM-1, soluble vascular cell adhesion molecule 1.

**Table 2 diagnostics-09-00028-t002:** Concentrations of plasma vitamin B6 forms according to the risk score of cardiovascular hospitalization in 122 patients with type 2 diabetes.

Plasma B6 Metabolite	Low risk ^1^ (<−2.8) *n* = 61	High risk ^1^ (≥−2.8) *n* = 61	*p* ^2^	*p* ^3^	*p* ^4^
4-Pyridoxic acid, nmol/L	31.9 (15.0) 27.3 [16.1]	48.2 (63.7) 36.2 [19.2]	0.031	0.160	0.166
4-Pyridoxic acid % of total vitamin B6	31 (10)	37 (9)	0.002	0.002	0.005
Pyridoxine, nmol/L	38.1 (127.8) 8.2 [7.0]	12.8 (28.4) 5.7 [8.5]	0.003	0.0107	0.010
Pyridoxine % of total vitamin B6	14.9 (18.3)	8.6 (8.7)	<0.001	0.008	0.011
Pyridoxal, nmol/L	13.9 (5.7) 12.6 [7.5]	15.6 (15.1) 12.0 [9.4]	0.539	0.921	0.851
Pyridoxal % of total vitamin B6	13.6 (5.3)	13.8 (6.4)	0.938	0.387	0.387
Pyridoxal 5′-phosphate, nmol/L	29.3 (16.2) 25.2 [22.4]	34.6 (35.0) 26.7 [20.6]	0.922	0.790	0.970
Pyridoxal 5′-phosphate % of total vitamin B6	28.3 (9.9)	26.2 (9.3)	0.220	0.397	0.658
Pyridoxamine, nmol/L	1.8 (1.0) 1.5 [1.3]	2.7 (3.5) 1.6 [1.9]	0.567	0.225	0.360
Pyridoxamine % of total vitamin B6	2.0 (1.3)	2.7 (3.5)	0.372	0.142	0.261
Pyridoxamine phosphate, nmol/L	10.0 (5.1) 9.9 [6.7]	11.5 (8.4) 10.1 [5.2]	0.619	0.517	0.731
Pyridoxamine phosphate % of total vitamin B6	10.3 (6.8)	11.4 (6.7)	0.419	0.342	0.583
Sum of vitamin B6 forms, nmol/L ^5^	132 (142) 98 [47]	126 (122) 96 [50]	0.967	0.309	0.378
4-Pyridoxic acid/pyridoxine ratio	3.9 (3.2) 3.1 [2.9]	7.0 (4.8) 5.8 [6.2]	<0.001	<0.001	0.001
PAr index ^6^	0.84 (0.06) 0.74 [0.43]	1.05 (0.07) 0.91 [0.64]	0.017	0.117	0.282

Data are mean (SD) and median [interquartile range]. ^1^ The risk score was calculated from classical risk factors using the formula in Appendix A, Appendix A [11]. ^2^
*p*-values are according to the Mann–Whitney test. ^3^
*p*-values are according to the univariate analysis of variance (ANOVA test) test applied on the log-transformed data and including cystatin C as a covariate. ^4^
*p*-values are according to the ANOVA test that included cystatin C and CRP as covariates. ^5^ The sum of plasma vitamin B6 forms (4-pyridoxic acid, pyridoxine, pyridoxal, pyridoxal 5′-phosphate, pyridoxamine, pyridoxamine phosphate). ^6^ PAr index = 4-pyridoxic acid/(pyridoxal + pyridoxal 5′-phosphate) ratio.

**Table 3 diagnostics-09-00028-t003:** Concentrations of urine vitamin B6 forms according to the risk score of cardiovascular hospitalization in 122 patients with type 2 diabetes.

Urine (U) B6 Metabolites	Low Risk ^1^ (<−2.8) *n* = 61	High risk ^1^ (≥−2.8) *n* = 61	*p* ^2^	*p* ^3^	*p* ^4^
U 4-Pyridoxic acid, µmol/g creatinine	4.4 (1.8) 4.0 [2.2]	5.3 (5.7) 3.9 [1.5]	0.840	0.395	0.348
U Pyridoxine, µmol/g creatinine	0.85 (1.17) 0.40 [0.98]	0.59 (0.87) 0.23 [0.58]	0.037	0.092	0.118
U Pyridoxal 5′-phosphate, µmol/g creatinine	0.16 (0.19) 0.07 [0.18]	0.13 (0.10) 0.10 [0.11]	0.772	0.201	0.249
U Pyridoxamine, µmol/g creatinine	0.06 (0.04) 0.04 [0.06]	0.07 (0.11) 0.05 [0.06]	0.680	0.328	0.358
U Pyridoxamine phosphate, µmol/g creatinine	0.49 (1.10) 0.13 [0.45]	0.59 (1.25) 0.21 [0.51]	0.308	0.704	0.807
U PAr index ^5^	13.9 (2.7)	16.0 (2.0)	0.041	0.080	0.080
U 4-Pyridoxic acid/pyridoxine ratio	27.0 (8.4)	37.5 (9.8)	0.022	0.068	0.084

Data are mean (SD) and median [interquartile range]. ^1^ The risk score was calculated from classical risk factors using the formula in Appendix A, Appendix A [11]. ^2^
*p*-values are according to the Mann–Whitney test. ^3^
*p*-values are according to the univariate analysis of variance (ANOVA test) test applied on the log-transformed data and including cystatin C as a covariate. ^4^
*p*-values are according to the ANOVA test that included cystatin C and CRP as covariates. ^5^ PAr index = 4-pyridoxic acid/ (pyridoxal + pyridoxal 5′-phosphate) ratio.

**Table 4 diagnostics-09-00028-t004:** Receiver operating curve to test the area under the curve (AUC) for candidate atherosclerosis prognostic markers in discriminating between diabetics according to cardiovascular risk scores.

Candidate Marker	AUC (95% CI) ^1^	*p*	Cut-Off (80% Sensitivity)	Corresponding 1-Specificity
4-Pyridoxic acid/pyridoxine ratio	0.72 (0.62–0.81)	<0.001	2.93	0.54
PAr index	0.63 (0.53–0.73)	0.017	0.60	0.73
sVCAM-1	0.68 (0.58–0.78)	0.001	397	0.61
Creatinine	0.63 (0.53–0.73)	0.013	0.75	0.71
Cystatin C	0.63 (0.53–0.73)	0.013	0.713	0.64
CRP	0.60 (0.50–0.70)	0.057	0.67	0.78

AUC (95% CI): area under the curve (95% confidence intervals). ^1^ Cardiovascular risk score according to Yu et al. [11] was entered as a binary outcome. 4-Pyridoxic acid/pyridoxine ratio and PAr index were determined in plasma, and sVCAM-1, creatinine, cystatin C, and CRP were measured in serum. CRP, C-reactive protein; sVCAM-1, soluble vascular cell adhesion molecule; PAr index = 4-pyridoxic acid/(pyridoxal + pyridoxal 5′-phosphate).

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
