# Peer review of "4-Pyridoxic Acid/Pyridoxine Ratio in Patients with Type 2 Diabetes is Related to Global Cardiovascular Risk Scores"

_diagnostics, 2019, doi:10.3390/diagnostics9010028_

Round 1

Reviewer 1 Report

This is a well-written article with clear presentation of study findings. The sample size however, is on a smaller side which may lead to us to question the power of this study. The authors are recommended to show a minimum sample size or power calculation to justify the number of samples. 

Author Response

Reply:

We agree with the reviewer. However, no sample size estimation was possible due to lack of similar studies using global vascular risks. This is mentioned now in the study limitations. Moreover, we now provided estimation of the study power with the sample size of 122 participants to detect differences in 4-pyridoxic acid/pyridoxine ratio between diabetics patients with high and low global risk.  

Reviewer 2 Report

Diagnostics 445996

Abstract: Please include a sentence mentioning the need of this study before the sentence “We investigated….” in the Background subsection.

Introduction: Please explain why the relationship between concentrations of vitamin B6 forms and global vascular risk deserve further investigations? Is there a lack of data or incidence and prevalence is increasing due to vitB6 deficiency? Authors have discussed the relationship between vitB6 and CV risk, but I can’t see the reason why this study is needed. There is a need to describe the correlation between disease, CV risks factors, and vitB6.

Study population and setting: Please mention how many total patients were recruited? What about the controls? In case of no controls, was the aim of the study was to correlate and compare between the different levels of blood sugars, or the B6 levels, or the CV risks score? Please mention it clearly or mention the research strategy. Please also mention that the clinical, medical, and personal history of the patients was acquired by medical record review.

“Framingham score applies up to the age of 75 years, thus it was not calculated from 29 patients who were older than 75 years.” Please either provide a table or flow chart of patients with age group, numbers, and gender. This sentence says 29 patients, how many were scored according to PROCAM (122-29=93 or 122)?

Please write down the inference at the end of each in the result section.

Vitamin B6 is metabolized in liver, please mention in the inclusion or exclusion criteria whether the patients under study had any type of liver disease? If yes, what were the results of LFT?

Overall this study proposes 4-pyridoxic acid/pyridoxine ratio as a marker based on an independent

 direct association with sVCAM-1 and global cardiovascular risk scores in diabetics. What about non-diabetics? Is there any study related? It will be interesting to see if the same marker can be used in non-diabetics or not?

My main concern is missing control and comparison with non-diabetics with CV risk/diseases. It will be interesting to discuss these issues in the discussion.

Author Response

Reviewer 2

Abstract: Please include a sentence mentioning   the need of this study before the sentence “We investigated….” in the   Background subsection.

Reply: We explained the need for the study in the abstract.  “Vascular diseases are multifactorial and   several risk factors may have synergetic effect on the global vascular risk.”

Introduction: Please explain why the relationship   between concentrations of vitamin B6 forms and global vascular risk deserve   further investigations? Is there a lack of data or incidence and prevalence   is increasing due to vitB6 deficiency? Authors have discussed the   relationship between vitB6 and CV risk, but I can’t see the reason why this   study is needed. There is a need to describe the correlation between disease,   CV risks factors, and vitB6.

Reply: We re-wrote parts of the introduction to explain the need for   the study:

“Elevated   PAr index is associated with a significant increase in the risk of future   endpoints such as stroke [17]   or mortality [2].   In the Hordaland Health Study, PAr has been shown to be a significant   predictor of the risk of stroke in strata of age, sex, body mass index (BMI),   smoking, hypertension, and diabetes [17].   Moreover, PAr was the strongest predictor of future stroke compared with each   of the classical risk factors (diabetes, smoking, hypertension, GFR, and CRP)   [17].   However, atherosclerosis is a multifactorial disease and the presence of   several risk factors in one person increase the risk of future vascular   events compared with the presence of any of the risk factors alone. It   is not clear if dysregulations of vitamin B6 metabolism (i.e., measured using   the surrogate marker PAr index) can be predicted by the presence of several   traditional vascular risk factors or with an increase of plasma   concentrations of early markers of atherosclerosis such as soluble vascular   cell adhesion molecule 1 (sVCAM-1).

In patients with type 2 diabetes who are at increased risk for   atherosclerosis, we hypothesized that dysregulation of vitamin B6 metabolism   shows a dose-response relationship with composite vascular risk scores that   are used as surrogate markers of future atherosclerosis risk.

Study population and setting: Please mention how   many total patients were recruited? What about the controls? In case of no   controls, was the aim of the study was to correlate and compare between the   different levels of blood sugars, or the B6 levels, or the CV risks score?   Please mention it clearly or mention the research strategy. Please also   mention that the clinical, medical, and personal history of the patients was   acquired by medical record review.

Reply: We added the two sentences:

122 patients with diabetes were   recruited.

Medical records were reviewed for disease history.   

“Framingham score applies up to the age of 75   years, thus it was not calculated from 29 patients who were older than 75   years.” Please either provide a table or flow chart of patients with age   group, numbers, and gender. This sentence says 29 patients, how many were   scored according to PROCAM (122-29=93 or 122)?

Reply: We now explained this. 93 were scored with Framingham and   PROCAM. A revised supplemental Figure 2 is now provided.

Please write down the inference at the end of   each in the result section.

Reply: We did as suggested

Vitamin B6 is metabolized in liver, please   mention in the inclusion or exclusion criteria whether the patients under   study had any type of liver disease? If yes, what were the results of LFT?

Reply: We did not perform clinical test to exclude patients with for   instance fatty liver. We measured ALT, alanine aminotransferase; and AST, aspartate   aminotransferase (shown in table 1). ALT, AST and the ratio of AST/ALT   (de-ritis index) are shown in Table 1. There was a significant difference in   the AST/ALT ratio according to global risk scores in patients with diabetes   (Table 1). However, ALT, AST and AST/ALT ratio showed no significant   correlation to PAr, 4-Pyridoxic   acid/pyridoxine ratio, or to any of the   vitamin B6 forms (except AST and pyridoxamine). Therefore, we do not think that liver function   is a confounding factor in the association between B6 forms and atherosclosis   scores in this study.     

Overall this study proposes 4-pyridoxic   acid/pyridoxine ratio as a marker based on an independent

 direct association with sVCAM-1 and global   cardiovascular risk scores in diabetics. What about non-diabetics? Is there   any study related? It will be interesting to see if the same marker can be   used in non-diabetics or not?

Reply: in the group of healthy subjects. We found   no correlation between PAr or 4-pyridoxic acid/pyridoxine ratio and sVCAM-1 or   global cardiovascular risk scores (PROCAM or Framingham).

The question of an association in a cohort with   multiple risk factors, but without diabetes worth further investigations, but   it cannot be answered in our study. 

My main concern is missing control and comparison   with non-diabetics with CV risk/diseases. It will be interesting to discuss   these issues in the discussion.

Reply: We agree that this control group is   interesting to test. However, we did not recruit patients with CVD risk   factors and without diabetes. We added this point to the study limitations.

Round 2

Reviewer 2 Report

Manuscript has been revised for most of the concerns. The limitations of the study has been added in discussion section. I would add limitation as a seperate subsection. 

Still the relationship and correlation with non-diabetic would be interested to investigate.

Author Response

we now have added a separate session on limitations of the study and added the open question suggested by the reviewer in the conclusion. The changes are marked in red.